# A Cognitive Method for Automatically Retrieving Complex Information on a Large Scale

**DOI:** 10.3390/s20113057

**Published:** 2020-05-28

**Authors:** Yongyue Wang, Beitong Yao, Tianbo Wang, Chunhe Xia, Xianghui Zhao

**Affiliations:** 1School of Computer Science and Engineering, Beijing Key Laboratory of Network Technology, Beihang University, Beijing 100191, China; buaawyy@buaa.edu.cn (Y.W.); yao6921@buaa.edu.cn (B.Y.); xch@buaa.edu.cn (C.X.); 2School of Cyber Science and Technology, Beihang University, Beijing 100191, China; wangtb@buaa.edu.cn; 3School of Computer Science and Information Technology, Guangxi Normal University, Guilin 541004, China; 4China Information Technology Security Evaluation Center, Beijing 100085, China

**Keywords:** information retriever sensor, multi-hop reasoning, evidence chains, complex search request

## Abstract

Modern retrieval systems tend to deteriorate because of their large output of useless and even misleading information, especially for complex search requests on a large scale. Complex information retrieval (IR) tasks requiring multi-hop reasoning need to fuse multiple scattered text across two or more documents. However, there are two challenges for multi-hop retrieval. To be specific, the first challenge is that since some important supporting facts have little lexical or semantic relationship with the retrieval query, the retriever often omits them; the second challenge is that once a retriever chooses misinformation related to the query as the entities of cognitive graphs, the retriever will fail. In this study, in order to improve the performance of retrievers in complex tasks, an intelligent sensor technique was proposed based on a sub-scope with cognitive reasoning (2SCR-IR), a novel method of retrieving reasoning paths over the cognitive graph to provide users with verified multi-hop reasoning chains. Inspired by the users’ process of step-by-step searching online, 2SCR-IR includes a dynamic fusion layer that starts from the entities mentioned in the given query, explores the cognitive graph dynamically built from the query and contexts, gradually finds relevant supporting entities mentioned in the given documents, and verifies the rationality of the retrieval facts. Our experimental results show that 2SCR-IR achieves competitive results on the HotpotQA full wiki and distractor settings, and outperforms the previous state-of-the-art methods by a more than two points absolute gain on the full wiki setting.

## 1. Introduction

Most previous work about retrievers focus on searching the relative lexicon from a single paragraph, and some easy search queries may be satisfied reluctantly using a single paragraph or document by the traditional retrieval methods of search engines, such as matching lexical or semantic similarity or relatedness. However, a very challenging retrieval task for complex search requests, called multi-hop reasoning retrieval, requires combining evidence from multiple sources, as shown in Figure 1. Single-hop retrieval methods are almost incompetent for intricate search tasks. The objective of a multi-hop retriever is to obtain and fuse scattered information from the internet step by step [1,2]. Figure 1 provides an example of a search request requiring multi-hop reasoning. In response to the search request, only when one first infers from the first context in the search request can the Internet information corresponding to the query be inferred with any confidence from the second context. 

The multi-hop IR is of great difficulty, because it requires reading and aggregating information over multiple pieces of textual evidence. Although the Reading Comprehension model is applied to candidates of the paragraph for prediction (e.g., QFE [3], MINIMAL [4]), the aggregation of evidence from sources for supporting fact prediction is beset with critical difficulty. One of the challenges is that since some pivotal supporting facts have little lexical or semantic relationship with the original retrieval query, the retriever often omits these crucial facts, as illustrated in Figure 2. The reason for this challenge is that multi-hop IR is open domain retrieval on a large scale, where search requests are given without any accompanying contexts. In this case, one is required to locate relevant contexts to queries from a large internet source before extracting the most relevant information. Retrieving a fixed list of documents independently does not capture the relationships between evidence documents through bridge entities required for multi-hop reasoning. They often fail to retrieve the required evidence for the multi-hop query. The recent open domain IR methods [5] do not capture lexical information in entities, but are encountered with challenges for entity-centric problems. The complex cognitive process in humans seems to provide more inspiration for this retrieval process. Recently, some researchers began to mimic the human reasoning process to search for complex information. Although Ding’s Cognitive Graph [6] takes the entity relations, this method cannot find the reasoning path automatically. Multi-Step Reasoner introduces a multi-step reasoning method [7], but it only repeats the retrieval process for a fixed number, resulting in ineffectively integrated information. Qi et al. propose GoldEn IR [8], which cannot use the information of relations between documents during the iterative process. Most state-of-the-art approaches [2,9,10,11] for the open-domain retrieval leverage non-parameterized models to retrieve a fixed set of documents, where an information span is extracted by a neural model. However, they often fail to retrieve the required evidence needed by multi-hop reasoning. Some research on the reasoning process do not separate the information from the cognitive graph. If helpful information for users is not selected on the cognitive graph, the retriever may fail. In addition, the explosion of online textual content with unknown verification raises a vital concern about misinformation, such as fake news, rumors, and online “water army” opinions [12,13,14]. While some misinformation may have semantic relations with the retrieval query, they cannot be efficaciously identified by some methods [15,16,17] with ideal performance. If the scattered information does not have stronger interactions with neighbors on the cognitive graph, a retriever is apt to select this misinformation in comparison with the whole nodes. While RE^3^ and Multi-Passage [11,18] introduce an extract-then-select framework to re-rank candidates to improve these interactions, they overlook the information between different evidence candidates.

In this paper, we study the problem of multi-hop retrieval methods, which requires multi-hop reasoning among evidence scattered around multiple raw documents on a large scale. Despite the given query utterance and a set of accompanying documents, not all of them are relevant. The information can be obtained by selecting two or more pieces of evidence from documents and reasoning among them. The models are expected to search the most useful information for search request in open domains. We propose a sub-scope cognitive reasoning information retrieval (2SCR-IR) sensor, a novel method to address the above-mentioned difficulties for multi-hop retrieving, as shown in Figure 3 (Nodes are fundamental entities, with the color denoting the identical entity in paragraphs. The blue edges are the relationship between the different entities in the same paragraph, while the red ones are the relationship between the same entities in different paragraphs. One search request and three paragraphs are given. Our 2SCR-IR sensor conducts multi-step reasoning over the supporting evidence by constructing a cognitive graph from the supporting information, automatically adjusting the sub-scope to select a subgraph, propagating information through the graph, verifying the reliability of the information, and finally producing supporting evidence chains. The circles, denoting the sub-scope, are chosen by 2SCR-IR in each step). For the first challenge, the 2SCR-IR sensor constructs a self-adjusting sub-scope cognitive graph based on entities referred to in the search request and web documents, which is iterated to accomplish multi-hop reasoning. In each step, the 2SCR-IR sensor is produced and reasons on a dynamically adjusted graph, with unrelated entities left out and reasoning information exclusively preserved in a scope prediction process. To solve the second difficult problem, we use an information fusion process in the 2SCR-IR sensor to eliminate misinformation. The process of iteratively expanding with clues in sub-scope can discover the weak paragraphs relative to the query in our framework, which can also play a pivotal role in filtering out misinformation. For the purpose of further verifying the authenticity of retrieved information, we introduce pageview verification to our sensor. Overall, our work incorporates a four-fold contribution:We proposed a 2SCR-IR sensor, a novel means for the multi-hop text-based retrieval tasks.We proposed a novel sub-scope module in the 2SCR-IR sensor capable of settling the problem that resulted from weak correlations between the queries and useful documents.To increase the retrieved information’s credibility, we introduce the fusion module and pageview information to our sensor to filter misinformation.Most IR methods using the “black box” merely provide users with retrieval results, generating utter ignorance of users about the logical relationship of the information they list. We proposed an explicit way to explain the reasoning process of the retriever. As a result, our 2SCR-IR sensor can produce reasoning chains, which can provide users with a range of structured documents or paragraphs. Compared with scattered and unstructured text, the reasoning chains can help users raise the efficiency of searching for complex information.

## 2. Related Works

### 2.1. Search Engine and Information Retrieval

The search engine is a system that assists users in retrieving information that they wish to obtain after submitting searching queries. One search engine, in response to a retrieval query, typically compares keywords of the query with an index generated from a sea of web sources, such as text files, image files, video files, and other content items. Based on this comparison, it will provide users with the most relevant content items [19]. In our paper, only textual information retrieved from the Internet is taken into account. With the rapid development of deep learning, researchers have an increasing interest in bringing neural networks into IR tasks. Although IR tasks frequently use sentence-similarity, as discussed by Guo et al. [20], they pay more attention to relevance-matching, in which the match of specific content plays a significant part. Some researchers [21] have confirmed that IR is more about retrieving sentences with the identical semantic meaning, while traditional methods based on relevance matching show a serious deficiency in semantic understanding. Therefore, we used natural language reasoning techniques, instead of relevance-focused IR methods, to solve this issue. In many cases, the paragraph containing the information corresponding to the searching query has great lexical overlap with the query, adding to its difficulty for the retrieval of common search engines from a large open scope. For instance, the accuracy of a BM25 retriever for finding all supporting evidence for a query diminishes from 57.3% to 25.9% on the “easy” and “hard” subsets, respectively [2]. 

### 2.2. Single-Hop and Multi-Hop QA

Relying on the complexity in underlying reasoning, information retrieval tasks can be categorized into single-hop and multi-hop ones. The former only needs one piece of evidence extracted from the underlying information, e.g., “which city is the capital of China”. On the contrary, a multi-hop retrieval task requires recognizing multiple relevant evidence and reasoning about them, e.g., “what is the capital city of the largest province in China”. Many IR techniques that are able to reply the single-hop searching query [22] are hardly introduced in multi-hop tasks, since single evidence can only partially satisfy its searching query. 

### 2.3. Open-Domain Retrieval Task

Open-Domain retrieval task refers to the setting where the search scope of the supporting evidence is extremely large and complex. Recent open domain IR systems based on deep learning methods follow a two-step process, namely selecting potentially relevant documents from a large corpus and extracting useful sources from these selected documents. As the first to successfully apply this framework to Chen et al. [23], IR [24], introduced new benchmarks, [11,25] improved this framework by introducing feedback signals, and [26] proposed a method that can dynamically adjust the number of retrieved documents. However, these mainstream methods are weak in mining the information loosely related to the searching queries and fail to identify the misinformation.

### 2.4. Similarity Matching Based Method

Traditional frequency-based approaches, such as the “bag of words” model [27], have been employed to extract features in the search engine. However, these frequency-based techniques do not preserve the text sequence, bringing about the lack of understanding of the context’s full meaning [28]. While other text-similarity-matching approaches, such as tree kernels [29] and high order n-grams [30], may have the perception of the word order and semantics, they cannot master the context meaning completely, thus heavily affecting the accuracy of recognition.

### 2.5. Multi-Hop Reasoning for Retrieval Task 

Previous research on popular GNN frameworks has shown promising results in retrieval tasks requiring multi-hop reasoning. Coref-GRN aggregates information in disparate references from scattered paragraphs [31]. The different mentions of the same entity are combined with a graph recurrent neural network (GRN) to produce aggregated entity representations [32]. Based on Coref-GRN, MHQA-GRN [33] refines the graph construction procedure with more connections, which shows further improvements. Entity-GRN [34] proposes a method to distinguish different relations in the cognitive graphs through the convolutional neural network. Besides, Core-GRN [35] explores the cognitive construction problem. Nonetheless, it is rarely investigated by researchers about how to effectively reason on the constructed cognitive graphs, which is the primary problem studied in this paper. Other frameworks, such as Memory Networks [36], deal with multi-hop reasoning. These frameworks develop representations for queries and retrieval sources, between which interactions are then made in a multi-hop manner. The IR-Net [37] generates an entity state and a relation state at each step, computing the similarity degree between all entities and relations given by the dataset. However, these frameworks perform reasoning on simple datasets with a limited number of entities and relations, which is quite different from our work with the large-scale retrieval and intricate search queries.

### 2.6. Pageview

The pageview is a collection of Web objects or resources representing a specific user event, such as clicking on a link or viewing a product page [38]. Besides, pageview frequency has been shown to help improve the performance of evidence retrieval results [39].

### 2.7. Peasoning Chains

A reasoning chain is a sequence that logically connect users’ search requests to supporting facts [40]. Reasoning chains should be intuitively related: they should exhibit a logical structure [41], or some other kind of textual relation that would allow human readers to quickly obtain what they really need.

## 3. System Architecture

### 3.1. Overview

In this section, we describe our new graph-based reasoning method that learns to find supporting evidence as reasoning paths for complex retrieval tasks. Our inspiration is motivated by the human cognitive process for searching an open question online, and our method starts from selecting the critical entity in the retrieval query. In order to make better use of the current entity’s surrounding information, our method is to connect the start entity to some related entities either searched in the neighborhood or connected by the same surface mentioned. The abovementioned process is repeated to construct reasoning chains.

The main part to implement the 2SCR-IR framework is comprised of two components in our proposed system (Figure 4): a system to construct the cognitive graph, on which the other system reasons explicitly. Our method uses a strong interconnection between evidence to prevent important documents from being omitted in the reasoning paths.

We used Wikipedia to retrieve relevant information for an open-domain search request, in which each article falls into paragraphs, resulting in billions of paragraphs in total. Any paragraph *p* can be perceived as our retrieval target. Given a query *q*, our framework aims at deriving its relevant textual sources to make up evidence chains by retrieving and reasoning paths, each of which is represented with a sequence of paragraphs. 

### 3.2. Gold Paragraph Selection

To ease the computational burden of following steps, we first applied paragraph selection to the given context. Documents with titles matching the whole query were our priority. The selector will calculate the relevance score of the query and each paragraph between 0 and 1. We can set a threshold parameter *τ*, and paragraphs with predicted scores greater than *τ* are chosen, which are concatenated together as the context C. Since not every piece of text is relevant to the query, we further search for paragraphs that contain entities appearing in the searching query. If the two previous ways fail, a BERT-based paragraph ranker [42] will be used to select the paragraph with the scores higher than *τ*. Q and C are further processed by upper layers. After the first-hop paragraphs are identified, the next step is to search for evidence within paragraphs leading to other relevant paragraphs. Unlike the dependence on entity linking, we use hyperlinks in the first-hop paragraphs to find second-hop paragraphs to avert introducing noise. After all the links are chosen, we add edges between the evidence with these links and hyperlinks. When the two-hop paragraph is selected, we can obtain some candidate paragraphs. In order to further lower noise, we use the paragraph ranking model based on the BERT encoder to select the top-*N* paragraphs in each step. Besides, a retrieval query *q* and each of the 10 paragraphs *p_i_* are introduced into a BERT model [42]. Furthermore, a soft-max function is utilized to calculate the probability *P* (*q*, *p_i_*) of *p_i_*. A paragraph *p_i_* is chosen as a gold paragraph for the query *q* if *P* (*q*, *p_i_*) is larger than the setting *τ*. To collect most of the gold paragraphs for each retrieval query, the threshold value of 97.0% for recall and 69.0% for precision was set.

### 3.3. Encoder and Attention Module

We introduced a BERT model to acquire the representation of the query Q=[q1,q2…qk]∈ℝk×d and that of the context C=[C1,…Cl]∈ℝl×d, where *k* and *l* are the lengths of the searching query and the context, and *d* is the size of the BERT hidden layer.

To gain more semantic information, following [43], we use the bidirectional attention module to update the representation for each word and a weighted self-aligned vector to represent the query.

The representation of the context *C* is
(1)C=[C;CQ;C⊙CQ;QC⊙CQ]∈ℝl×4dh
where ⊙ is elementwise multiplication, and dh is the hidden dimension.

The representation of the searching query is
(2)$$\varphi =softmax\left(w_{\varphi }h\right)$$
(3)$$q=w_{a}tanh\left(W_{a}\underset{k}{\sum }\varphi_{k}h_{k}\right)$$
where *h* represents the query embedding; $h_{k}$ denotes the *k*-th word in the query; and $w_{\varphi }$, $w_{a}$, and $W_{a}$ are linear projection parameters.

### 3.4. Construction of Cognitive Graph

To perform the multi-hop reasoning, we first needed to construct a graph of the information source to cover all the Wikipedia paragraphs. The Stanford CoreNLP Toolkit [44] was employed in an attempt to generate semantic graphs from the source text. The cognitive graph was constructed with entities as nodes and edges, while the number of extracted entities was denoted as *N*. We constructed the entity graph based on Entity-GCN [34] and DFGN [16], signifying that all mentions of the candidates searched in the documents are used as nodes in this cognitive graph. We used hyperlinks in Wikipedia to develop the direct edges, while the undirected edges were defined in line with the positional properties of every pair of nodes. The edges in this graph were split into two categories:The cross-document edge for every pair of nodes with the same entity located in different documents.The within-document edge for every pair of nodes located in the same document, including sentence-level, paragraph-level, and context-level links.

It needs to be emphasized that we do not apply the co-reference resolution to pronouns because it will introduce complex and unnecessary links. 

### 3.5. Multi-Step Reasoning

After context encoding, the 2SCR-IR sensor performs reasoning over the cognitive graph. With the embedding process of the query Q and the context C, there is a huge challenge: to identify support entities and the text span of potential evidence paragraphs, as well as capturing relationships between evidence documents with little lexical overlap or semantic relationship to the original query. Based on the Cognitive Graph [6] and the DFGN method [16] for the QA system, an advanced knowledge-fusion module is adopted to mimic humans’ step-by-step thinking and reasoning behavior—starting from $Q_{0}$ as well as $C_{0}$ and finding one-step supporting entities. The module achieves the following:
passing information from tokens to entities (Doc2Graph flow);propagating information on the entity graph;passing information from the entity graph to tokens (Graph2Doc flow).

After obtaining the entity embeddings from the input context $C_{t}$, we applied a graph neural network to propagate the node information to its neighbors. Furthermore, a dynamic graph attention mechanism, which is a hierarchical and top-down process, was utilized to imitate humans’ step-by-step exploring and reasoning behavior. It first attentively reads all knowledge graphs and then all triples in each graph for the final word generation. Yan et al. [45] show that the higher relevance given to the query can help the neighbor entity obtain more information from nearby. In order to calculate the degree of relation between the context and the query, 2SCR-IR calculates the attention between them. If one entity is more relative to the query, the entity is more pivotal. We followed [16] to multiply this relevance score:(4)$$\gamma_{i}^{\left(t\right)}=sigmoid\left(\frac{{\tilde{q}}^{\left(t-1\right)}V^{\left(t\right)}e_{i}^{\left(t-1\right)}}{\sqrt{d_{2}}}\right)$$
(5)$${\tilde{E}}^{\left(t-1\right)}=\gamma^{\left(t\right)}⊙e^{\left(t-1\right)}$$
where $V^{\left(t\right)}$ is a linear projection matrix; $d_{2}$ represents the dimension of each entity; and $\gamma_{i}^{\left(t\right)}$ is the relevance score of the entity $e_{i}$ in step *t*. The entity graph is updated in a loop.

As a result, this step of the information propagation is confined to a dynamic scope on the cognitive graph. The next step is to disseminate information across the dynamic scope. 

We introduce how to integrate the output of the entity into the context. To identify which sentences are more crucial to answer the question, we used a weighted self-aligned vector to represent the context. For each sentence $s_{i}$, we compute weight $\varepsilon_{i}$ by
(6)$$\varepsilon_{i}=softmax\left(w_{\varphi }s_{i}\right)$$

The context vector is computed by
(7)CV=watanh(Wa∑iεisi)
where wφ, wa are linear projection parameters, the context vector CV∈ℝd. 

After the context vector is concatenated to each word in the context, the *i*-th word $c_{i}$ in the context C is updated as follows:(8)$$c_{i}=\left[c_{i};C_{V}\right]$$

After completing the contextual information, we used an LSTM layer to fuse the output of the entity graph and the context:(9)C=LSTM([C;E])

The structure layer we used is the same as that used by Yang et al. [2]. This prediction process has three functions, including support sentences as well as the start and end position of the information needed, denoted as $O_{sup}$, $\;O_{start}$, $O_{end},$ respectively. We put three RNNs $P_{i}$ layer by layer to solve the problem of output dependency. The information of the fusion block is sent to the first RNN $P_{0}$, and the output of this module is as follows:(10)$$O_{sup}=P_{0}\left(C^{\left(t\right)}\right)$$
(11)$$O_{start}=P_{1}\left(\left[C^{\left(t\right)},O_{sup}\right]\right)$$
(12)$$O_{end}=P_{2}\left(\left[C^{\left(t\right)},O_{sup},\;O_{end}\right]\right)$$

Each Pi outputs a logit O∈ℝM×d2.

In order to optimize the combined effect of these three sections, we compute these three cross entropy losses:(13)$$L=\mu_{start}L_{start}+\mu_{end}L_{end}+\mu_{spport}L_{support}$$

Our method learns to extract reasoning paths within the constructed cognitive graph as evidence chains. Besides, evidence resources for a complex searching query do not necessarily have lexical overlaps. We adopted an automatically adjusting sub-scope to collect those entities with some sort of relation. Once one of the resources is retrieved, its entity mentions and the query often entail another resource.

### 3.6. Producing Plausible Evidence Chains

Finally, the 2SCR-IR sensor will provide users with reasoning chains, which first verifies every reasoning path, and then extracts a retrieval span from the most reasonable paths using a common method [42]. Our retriever model learns to predict plausible reasoning paths by capturing the paragraph interactions through the BERT (CLS) representations, after independently encoding the paragraphs along with the searching query. This paragraph interaction is crucial for multi-hop reasoning [46], especially when faced with open-domain information. However, as there is invariably some noise or misinformation, the best evidence chain is not always enough to fully cover all information to be retrieved. Hence, we re-computed the probability of the path with the increase in the pageview to lower the uncertainty and make our framework more robust. In the end, we select the top-*m* evidence chains for users. 

### 3.7. Example

After *N* paragraphs and the query are put in, the selector module will filter out irrelevant paragraphs. Then, BERT is used as the encoder module to represent the filtered content in line with Formulas (1)–(3). After the process of knowledge representation, a classification layer is applied to the prediction of relevance scores between the paragraphs and the query. One paragraph will be signed as 1 if it contains supporting facts. Otherwise, it will be signed as 0. In the inference process, the paragraph will be chosen if the relevance score exceeds *τ*. In the process of reasoning, in order to make inferences on information, we calculated the degree of the relationship between the entities and the query in the light of Formulas (4) and (5) and built the entity graph. After we got the output of the entity graph, it was fused into the context. We used Formulas (6)–(9) to recognize the relevance of the search request to sentences. Next, Formulas (10)–(12) were utilized to predict the supporting sentences, as well as the start and end position corresponding to each search request. Moreover, Formula (13) was employed to optimize the combined effect. Finally, we will get supporting facts and construct reasoning chains in correspondence with the search request. 

## 4. Experiments

### 4.1. Datasets

We evaluate our method on an open-domain Wikipedia-sourced dataset named HotpotQA [2], which is a human-annotated large-scale dataset that needs multi-hop reasoning and provides annotations to evaluate the prediction of supporting sentences. Almost 84% of queries require multi-reasoning. We used the full-wiki and the distractor setting of HotpotQA to conduct our experiments, with our primary target being the full wiki setting. Due to its open-domain scenario, we used the distractor setting to evaluate the performance of our method in a closed scenario where the evidence candidates are provided. For the sake of optimization, we used the Adam Optimizer [47] with an initial learning rate of 0.0005 and a mini-batch size of 32. 

In addition, a good information retriever should be robust enough to prevent noise from disturbing themselves. IR can be armed with the well-structured and large-scale knowledge graph DBPedia [48] to offer some robust knowledge between concepts. However, the biggest disadvantage of this approach lies in that our cognitive model cannot be trained end-to-end and the errors may be cascaded. Inspired by [23], our retriever is trained to recognize relevant and irrelevant paragraphs at each step. We therefore mixed the noise information (negative examples) and generally right paragraphs together; to be more specific, we used two types of negative examples: Term Frequency–Inverse Document Frequency-based (TF–IDF-based) and hyperlink-based ones. When it comes to the single-hop retriever, merely the former type is used. Regarding the multi-hop retriever, we used both sorts, with the latter type carrying more weight to prevent our retriever from being distracted by reasoning paths without correct answer spans. Generally, the number of negative examples is set to 50. 

### 4.2. Implementation Details

We performed our experiment in the Co-lab. In the section paragraph sector, the setting of the threshold τ is 0.1, with the length of the context restricted to 512. Besides, the number of entities in the cognitive graph was set to be 100. The input dimension of the BERT, the hidden dimension, and the batch size were set to be 800, 310, and 10, respectively. The total train epoch was set to be 30 and the first 5 epochs.

### 4.3. Baseline

We used Yang’s model [2] proposed in the original HotpotQA paper as a baseline model that follows the retrieval–extraction framework of DrQA [23] and introduces the advanced techniques, such as self-attention and bi-attention.

### 4.4. Evaluation Metrics 

First, we introduced two common metrics to evaluation metrics: the Exact Match (EM) and the F1 score [2]. For better a performance evaluation of the multi-hop reasoning concerning our sensor, we also introduced the supporting fact retrieval based on the common metrics. We adopted Supporting fact Prediction F1 (SPF1) and Supporting fact Prediction EM (SPEM) to evaluate the sentence-level supporting fact retrieval accuracy. It is worth noting that the paragraph-level retrieval accuracy matters for the multi-hop reasoning as well. Thus, we adopted Paragraph Recall (PR), which evaluates if at least one of the ground-truth paragraphs is included among the retrieved paragraphs. To evaluate whether both of the ground-truth paragraphs used for multi-hop reasoning are included among the retrieved paragraphs, we put Paragraph Exact Match (PEM) into use.

## 5. Results

### 5.1. Overall Results

Table 1 shows the performance of dissimilar IR models on the HotpotQA development set. From the table, we can see that the 2SCR-IR sensor outperforms all previous results under both the full wiki and distractor settings. Compared with the state-of-the-art model [15], 2SCR-IR achieves 1.1 F1 and 1 EM gains on the full wiki, as well as 1.2 F1 and 1.9 EM gains on the distractor wiki. The result shows that 2SCR-IR achieves improvement in predicting supporting facts both in the full wiki and distractor wiki settings. 

After comparing our 2SCR-IR sensor with other competitive retrieval methods on SQuAD datasets, our model is found to outperform the current state-of-the-art model [49] by 2.6 F1 and 3 EM scores, as shown in Table 2 At the beginning of the paper, we predicted that the lower lexical overlap between the search query and contexts will pose a challenge to the methods using lexical-based retrievers in finding relevant articles. The experiment proved that our prediction is valid.

### 5.2. Performance of Evidence Chain Retrieval

Retrieval results in Table 3 display that our 2SCR-IR sensor generates the improvement of 1.3 PR, 7.2 PEM, and 7.8 EM, respectively. The conspicuous improvement from TF-IDF to Entity-centric IR demonstrates that exploring different granularity reasoning helps to retrieve the paragraphs with fewer term overlaps. Moreover, the comparison of our retriever with Entity-centric IR Retrieval shows the importance of explicitly retrieved reasoning paths in the cognitive graph, especially for the complex multi-hop searching query. 

### 5.3. Ablation Study

To evaluate the performance of the disparate components in our 2SCR-IR sensor, we performed ablation studies, where we simply use golden supporting facts as the input context. Table 2 shows the ablation results of the multi-hop retrieval performances in the development set of HotpotQA. 

From Table 4, we can observe that the sub-scope and pageview parts can help our sensor obtain from 3 to 7% relative and 5 to 8% gains for F1 and EM, respectively. ‘‘-attention module’’ means discarding the information of bidirectional attention. Similarly, the performance drops a lot in each metric, which proves the capability of the bidirectional module to comprehend semantics.

At the beginning of the paper, we predicted that the pageview and the sub-scope can improve the ability to find useful supporting facts. In the experiment, our observation and proposed scheme are proved valid.

## 6. Case Study 

Our case study is presented in Figure 5, which signifies the reasoning process with a 2SCR-IR sensor. Firstly, our model produces scope 1 as the first entity of reasoning by comparing the searching query with entities, in which “Ran Paul” and “hotel” are selected as the first entities of two reasoning chains, the information of which is then passed to their neighbors on the cognitive graph. Secondly, mentions of the same entity “Galt house” are detected by scope 2, serving as a bridge for propagating information across two paragraphs. Thirdly, these two reasoning chains are linked together by the bridge entity “Galt house”. Both reasoning chains and supporting facts are provided to users, which will assist them in obtaining things they want to retrieve very promptly and efficiently. Finally, users can acquire “The Ran Paul presidential campaign 2016 event was held at a hotel which is on the Ohio River”, nearly without any complex manual retrieval process.

Figure 6 shows another multi-hop reasoning example of 2SCR-IR from the HotpotQA development set. In this case, we need to search information about the actress who played *Corliss Archer* in *A Kiss for Corliss*. The information relevant to the search request may appear in multiple positions in texts, but they have little lexical or semantic relationship to the original retrieval query. It is generally difficult for common search engine system to know which entity mentions might eventually lead to texts containing what the user really needs. Our 2SCR-IR can provide users with a logically structured text. From the reasoning chain and three supporting facts, we can easily get the information: *The woman who portrayed Corliss Archer in the film A Kiss for Corliss held Chief of Protocol in the government*. As shown in Figure 7, our 2SCR-IR method has been successfully applied to a KLBNT Retrieval System. For complex problems, we can get the reasoning process and the logically structured documents.

## 7. Conclusions

We present a new framework 2SCR-IR sensor to tackle multi-hop retrieval problems on a large scale, which retrieves reasoning paths over the cognitive graph to provide users with useful explicit evidence chains. Our retriever model learns to sequentially retrieve evidence paragraphs to construct reasoning paths, which is subsequently re-ranked by the sensor that determines the final information presented as the one extracted from the best reasoning path. Our retriever obtains state-of-the-art results using the HotpotQA dataset, which shows the efficiency of our framework. The state-of-the-art performance on SQuAD is achieved, demonstrating the robustness of our method. Besides, our analysis shows that 2SCR-IR can produce reliable and explainable reasoning chains. In the future, we may incorporate new advances in building cognitive graphs from the web context to solve more difficult reasoning problems. 

## Figures and Tables

**Figure 1 sensors-20-03057-f001:**
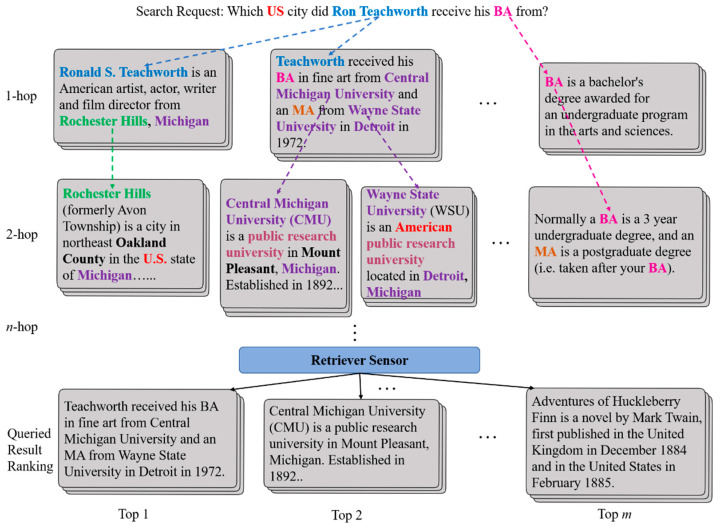
An example of a multi-hop retrieval task. The method in this paper mimics the human reasoning process while searching for information online. The retrieval sensor will finally provide users with reasoning chains, which will help obtain structured retrieval information.

**Figure 2 sensors-20-03057-f002:**
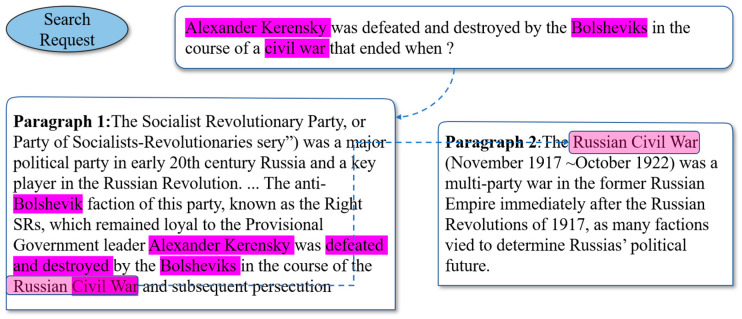
A complex example of an open-domain retrieval task from HotpotQA. Paragraph 2 is difficult to be retrieved using traditional retrievers due to little lexical overlap with the given query.

**Figure 3 sensors-20-03057-f003:**
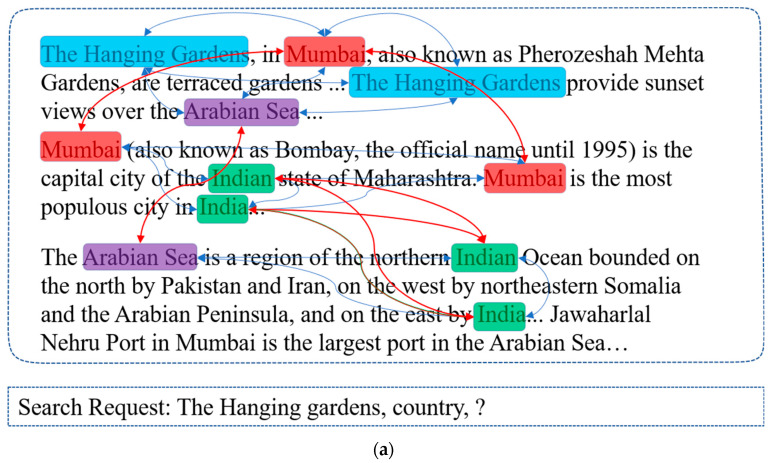
An example of a multi-hop text-based information retrieval task. (**a**). Search request and corresponding information source. (**b**). Reasoning process.

**Figure 4 sensors-20-03057-f004:**
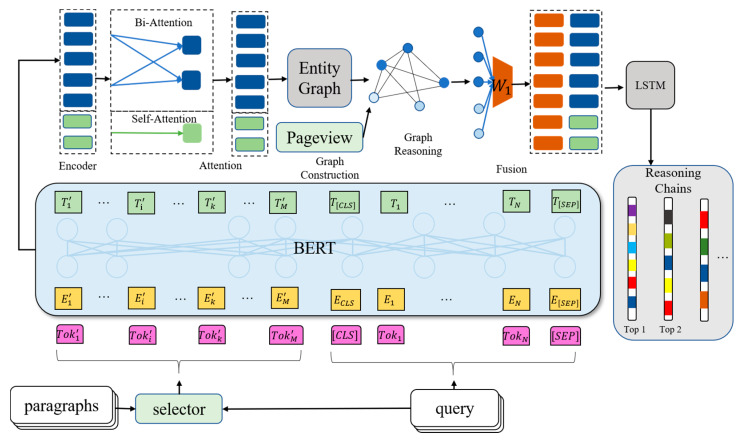
Model architecture of the proposed 2SCR-IR sensor.

**Figure 5 sensors-20-03057-f005:**
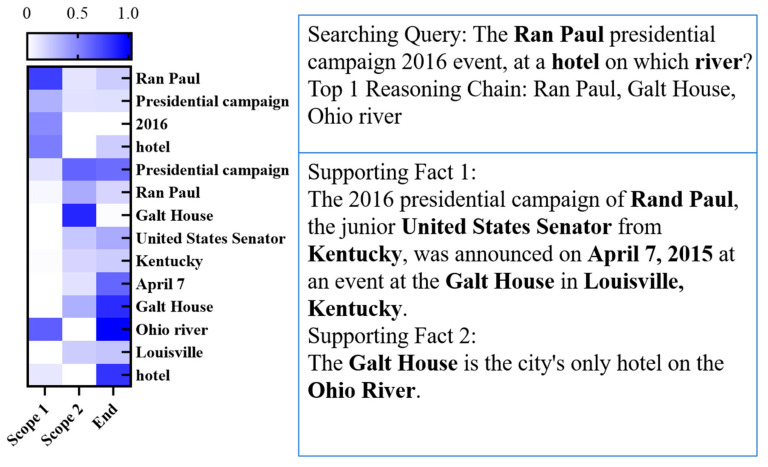
A case study of the development set. The number on the left side shows the importance scores of the predicted sub-scope. The text on the right side includes queries, predicted top-1 reasoning chains, and supporting facts.

**Figure 6 sensors-20-03057-f006:**
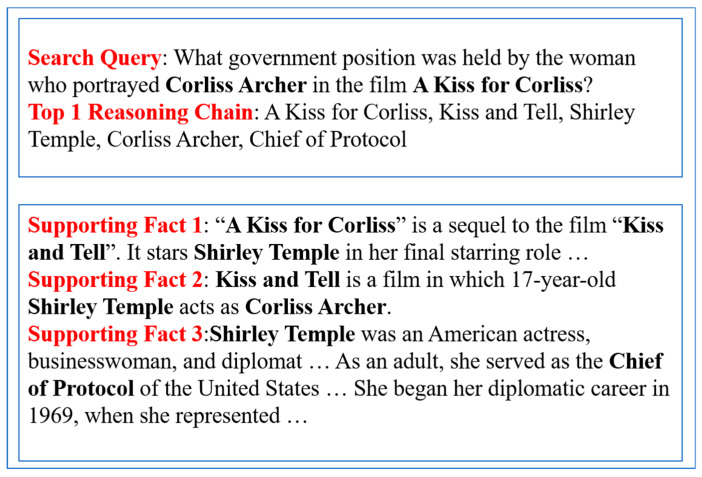
Another case study in the development set. Several documents are given as supporting facts for the search query.

**Figure 7 sensors-20-03057-f007:**
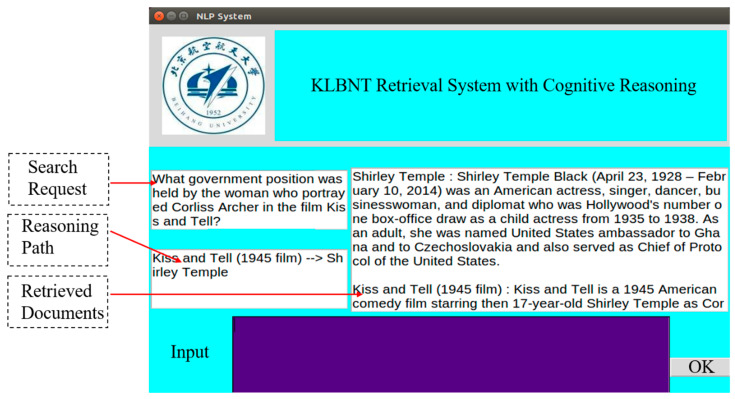
The KLBNT Retrieval System supported by 2RCR-IR technology.

**Table 1 sensors-20-03057-t001:** Primary results for HotpotQA development set results: SP results on the HotpotQA’s full wiki and distractor settings. The highest value per column is marked in bold.

Models	Full-Wiki SP		Distractor SP	
	F1	EM	F1	EM
Baseline [2]	40.7	5.4	68.3	23.2
MUPPET [7]	47.7	16.5	76.1	45.6
Cognitive Graph [6]	57.2	24.6	60.3	38.2
GoldEn Retriever [8]	65.2	31.4	69.8	45.4
Semantic Retrieval [9]	72.3	41.2	73.2	50.7
DFGN [16]	73.2	43.7	85.1	58.3
QFE [3]	45.2	13.9	82.9	56.7
Asai [15]	76.1	49.2	85.1	59.3
**Our method**	**77.2**	**50.2**	**86.3**	**61.2**

**Table 2 sensors-20-03057-t002:** SQuAD results: We report the F1 and EM scores on SQuAD, following previous work. The highest value per column is marked in bold.

Models	F1	EM
R^3^ [11]	37.8	28.9
Multi-step Reasoner [41]	39.3	32.7
MINIMAL [4]	42.6	34.9
DENSPI-hybrid [5]	45.1	36.7
BERTserini [9]	45.9	38.4
MUPPET [7]	46.3	39.3
RE^3^ [18]	51.2	42.3
multi-passage [49]	61.6	54.1
**Our method**	**64.2**	**57.1**

**Table 3 sensors-20-03057-t003:** Retrieval evaluation: Comparing our retrieval method with other methods across Paragraph Recall, Paragraph EM, and EM metrics. The highest value per column is marked in bold.

Methods	Paragraph Recall (PR)	Paragraph EM (PEM)	EM
TF-IDF [23]	66.7	9.8	18.3
Re-rank [50]	86.2	28.9	36.1
Entity-centric IR [17]	87.2	35.3	41.6
Cognitive Graph [6]	88.2	56.9	38.4
Semantic Retrieval [39]	92.9	65.7	46.3
**Our method**	**94.2**	**72.9**	**54.1**

**Table 4 sensors-20-03057-t004:** Ablation results on dev set.

Model	F1	EM
2SCR-IR	86.3	61.2
-sub-scope and pageview part	78.6	56.2
-attention module	82.7	53.9

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
