# Peer review of "A Cognitive Method for Automatically Retrieving Complex Information on a Large Scale"

_sensors, 2020, doi:10.3390/s20113057_

Round 1

Reviewer 1 Report

The paper describes a framework able to retrieve information on a large scale open domain, using multi-hop strategy. It pleasantly mimics human reasoning.

Below are my comments on the paper

  1. English should be improved by a native-english speaker.
  2. The introduction is relatively clear but the method should be described in more details, with more examples. Authors use many methods to which they refer (BERT-based paragraph ranker, Entity-GCN, DFGN,  ...). They all should be briefly described.
  3. Formulas and objects of all subsections of section 3 (System architecture) should all be described with examples. Formulas are given but it's really hard to follow what all objects refer without a clear example. I would recommend authors to complete their paper with an example (maybe in a additional section) in which they provide the results of all formulas and provide examples of used objects. It is not clear how linguistic, semantic and contextual aspects of the search are dealt with
  4. Authors shoud discuss why they did not use the DBpedia ontology to support their search and provide some robust knowledge between concepts.

Once all these comment have been taken into account, the paper may be ready to be published.

Reviewer 2 Report

The paper proposes a graph-based information retrieval technique for multi-hop reasoning. The approach is tested on publicly available datasets (HotpotQA and SQuAD) and the results show that the approach outperforms the baselines. Generally, the paper is well written and easy to follow. Nevertheless, I believe there are some issues that need to be addressed before the paper is publishable.

- My main concern is the novelty of the approach. To me it seems that the authors combined different already existing techniques. The contribution should be made more clear with respect to existing works and with respect to the "building blocks" on which the work is based.

- I am a bit confused about the choice of title. Why "cognitive sensor"? Why "sensor" at all when you basically have a IR method? And why "cognitive"? Here one would expect some reference to cognitive science and decision making, which I did not see in the paper.

- There are various approaches for generating semantic graphs from texts. Maybe the authors should look at such works and discuss how their approach could benefit from such approaches. For example, look at the Text2HBM project (text2hbm.org), where the goal is to generate "situation models" out of textual sources. Also, I could imagine that language taxonomies like WordNet could be helpful for enhancing the approach and its "explainability".

- The authors claim that their approach can produce "reliable and explainable reasoning chains". I would have liked to see more evidence both in terms of reliable chains and explainable chains. How exactly do the authors measure reliability and explainability?

- In the results section, the authors claim that their approach significantly outperforms existing models (Table 1). How did the authors measure the significance? Looking at the table I think that at least compared to Asai, the method is not significantly better. Here some statistical tests would be useful.

- In table 1, there is a problem with the caption. I think the authors did not remove the standard text from the latex template.

- At places there is problem with the English language (mostly grammar and some wording). Some proof reading should be done.

Author Response

Dear reviewer,

We have carefully read your comments one by one and revised our paper. And the revised paper are put in the attachment. Thank you very much for your valuable suggestions.

Yours Sincerely,

Yongyue Wang

  1. My main concern is the novelty of the approach. To me it seems that the authors combined different already existing techniques. The contribution should be made clearer with respect to existing works and with respect to the "building blocks" on which the work is based.

It is our fault that we did not clearly describe the relations between the previous work and our contributions. We really draw lessons from many advanced IR techniques. However, the problem we have solved is quite different from previous work. We combine some of the existing techniques to build a new model and solve the new problem of weak correlation between query and important information. I have improved our contribution in the section 1. (See lines 49, 50, 51, 52, 60, 61, 62, 63, 64, 65, 66, 67, 74, 75, 76, 107, 108 and 109 for details)

  1. I am a bit confused about the choice of title. Why "cognitive sensor"? Why "sensor" at all when you basically have a IR method? And why "cognitive"? Here one would expect some reference to cognitive science and decision making, which I did not see in the paper.

The title of this paper is intended to describe a virtual sensor with the function of retrieving complex information. Thank you for your suggestion. I’d like to change the title to “A Cognitive Method for Automatically Retrieving Complex Information on a Large Scale”. As for “cognitive”, our IR method mimic human’s step-by-step thinking and reasoning behavior when they search a complex information online. Since we don’t have a clear description, it is easy to confuse the readers. We have added the relevant description. (See lines 194, 195, 275, 276 and 277 for details)

  1. There are various approaches for generating semantic graphs from texts. Maybe the authors should look at such works and discuss how their approach could benefit from such approaches. For example, look at the Text2HBM project (text2hbm.org), where the goal is to generate "situation models" out of textual sources. Also, I could imagine that language taxonomies like WordNet could be helpful for enhancing the approach and its "explainability".

In order to be consistent with other methods in terms of experimental conditions, our method uses the Stanford Corenlp Toolkit[1] to recognize the entities and relations between them to support cognitive graph construction. Due to my negligence, I forgot to write this important factor in my paper, and I have added it to our paper in section 3.4. In addition, the reviewer provides us with an exciting suggestion. Maybe we can further break through the bottleneck of IR by improving the construction of cognitive graph. (See lines 248 and 249 for details)

  1. The authors claim that their approach can produce "reliable and explainable reasoning chains". I would have liked to see more evidence both in terms of reliable chains and explainable chains. How exactly do the authors measure reliability and explainability?

Most IR models use “black boxes”, showing only the search request and retrieval results. Our method provides the supporting facts and reasoning chains. Users can interpret the process more exhaustive with step by step solutions. In other words, previous methods have not provided an explicit and complete interpretation of retrieval process. Explicit reasoning paths are clear and reliable to explain the reasoning process. I am so sorry that we did not make it clear in our paper, and the corresponding content has been added in introduction. In order to make it clear for readers, we add another case study and a screenshot of the retrieval system we developed (figure 6 and figure 7). (See lines 186, 187, 188, 190 and 451 to 461 for details)

  1. In the results section, the authors claim that their approach significantly outperforms existing models (Table 1). How did the authors measure the significance? Looking at the table I think that at least compared to Asai, the method is not significantly better. Here some statistical tests would be useful.

Because of many reasons (such as poor quality of datasets), the performance of deep learning methods has been very difficult to be improved when reaching a certain level. However, after careful consideration, I feel that it is really not rigorous to use “significantly’ and I have corrected this defect. (See lines 397, 400 and 401 for details)

  1. In table 1, there is a problem with the caption. I think the authors did not remove the standard text from the latex template.

Sorry for this error due to our negligence, and we have corrected it. (See lines 402 and 403)

  1. At places there is problem with the English language (mostly grammar and some wording). Some proof reading should be done.

I am so sorry that I did not describe the paper clearly. We have invited a native-English teacher to provide a professional English editing service and finished the proof reading. All the changes have been marked.

The above questions raised by the reviewer are professional and of high quality, which are very helpful for our paper. Thanks again to the reviewer for his/her valuable and helpful comments!!!

[1]          C. D. Manning, M. Surdeanu, J. Bauer, J. R. Finkel, S. Bethard, and D. McClosky, "The Stanford CoreNLP natural language processing toolkit," in Proceedings of 52nd annual meeting of the association for computational linguistics: system demonstrations, 2014, pp. 55-60.

Round 2

Reviewer 2 Report

I believe that the authors have addressed all my comments. Maybe to improve the expectations from the title and the content of the paper, it would be useful if the authors mention the motivation behind calling their method "cognitive" early in the introduction.

A minor formatting problem is that at places there are no spaces between references and text. Also check again about typos.

Author Response

Thank you again for your valuable suggestions on my manuscript. Your suggestions are of great help to me.

1、It would be useful if the authors mention the motivation behind calling their method "cognitive" early in the introduction.

That is good advice. We have added brief sentences about the motivation early in the introduction. (As shown in lines 62, 63 and 64)

2、A minor formatting problem is that at places there are no spaces between references and text. Also check again about typos.

Thank you very much!!! we have corrected them.